# USP37 Deubiquitinates CDC73 in HPT-JT Syndrome

**DOI:** 10.3390/ijms23126364

**Published:** 2022-06-07

**Authors:** Su Yeon Kim, Ji-young Lee, Yun-jung Cho, Kwan Hoon Jo, Eun Sook Kim, Je Ho Han, Kwang-Hyun Baek, Sung-dae Moon

**Affiliations:** 1Institute of Biomedical Industry, College of Medicine, The Catholic University of Korea, Seoul 06591, Korea; sy5135sy@naver.com (S.Y.K.); jiyoung8889@naver.com (J.-y.L.); 2Division of Endocrinology and Metabolism, Department of Internal Medicine, Incheon St. Mary’s Hospital, College of Medicine, The Catholic University of Korea, Incheon 21431, Korea; deity393@naver.com (Y.-j.C.); lovi@naver.com (K.H.J.); ab13900@gmail.com (E.S.K.); jehohan@gmail.com (J.H.H.); 3Department of Biomedical Science, CHA University, Seongnam 13488, Korea; baek@cha.ac.kr

**Keywords:** hyperparathyroidism, cell division cycle, ubiquitination, deubiquitinating enzyme, ubiquitin-specific proteases, ubiquitin–proteasome system

## Abstract

The *CDC73/HRPT2* gene, a defect which causes hyperparathyroidism–jaw tumor (HPT-JT) syndrome, encodes CDC73/parafibromin. We aimed to investigate whether CDC73 would be a target for ubiquitin–proteasome degradation. We cloned full-length cDNAs encoding a family of 58 ubiquitin-specific deubiquitinating enzymes (DUBs), also known as ubiquitin-specific proteases (USPs). Use of the yeast two-hybrid system then enabled us to identify USP37 as interacting with CDC73. The biochemical interaction between the USP37 and CDC73 and their reciprocal binding domains were studied. Co-localization of CDC73 and USP37 was observed in cells. CDC73 was found to be polyubiquitinated, and polyubiquitination of CDC73 was prominent in mutants. CDC73 was deubiquitinated via K48-specific ubiquitin chains by USP37, but not by the catalytically inactive USP37^C350S^ mutant. Observation of the binding between deletion mutants of CDC73 and USP37 revealed that the β-catenin binding site of CDC73 and the ubiquitin-interacting motifs 2 and 3 (UIM2 and 3) of USP37 were responsible for the interaction between the two proteins. Moreover, these two enzymes co-existed within the nucleus of COS7 cells. We conclude that USP37 is a DUB for CDC73 and that the two proteins interact through specific domains, suggesting that USP37 is responsible for the stability of CDC73 in HPT-JT syndrome.

## 1. Introduction

Mutations in *CDC73* are associated with hyperparathyroidism–jaw tumor (HPT-JT) syndrome, an autosomal dominant disease characterized by parathyroid tumors, fibro-osseous jaw tumors, cystic kidney lesions and uterine tumors [1,2,3,4].

Human CDC73, also known as parafibromin, is encoded by the cell division cycle 73 gene (*CDC73*, also known as *HRPT2*) located on chromosome 1q31.2 [1,2,3]. CDC73 is a nuclear protein that interacts with β-catenin and forms the RNA polymerase-associated factor 1 complex (Paf1C) that regulates transcription [1,2,4,5,6,7]. The evolutionarily conserved Paf1C consists of five subunits (CDC73, Paf1, Ctr9, Leo1 and Rtf1, in yeast), binds RNA polymerase II, plays an important role in gene regulation and is implicated in development and human disease [8,9]. CDC73 was initially proposed to be a tumor suppressor that repressed the expression of cyclin D1 [1,10,11] and the c-myc proto-oncogene [1,4,10,12]. In other cases, it has oncogenic properties and interacts directly with nuclear β-catenin to act as a transcriptional co-activator of the Wnt/β-catenin signaling pathway [1,13]. Both the tumor-suppressive and oncogenic effects of CDC73 rely on the N-terminal portion of CDC73 [1,11,13].

The role of *CDC73/HRPT2* and the encoded CDC73/parafibromin protein in endocrine tumors has not been fully elucidated [1,2,14]. *CDC73* mutations associated with premature truncation have been predicted to lose key functional domains, including interaction domains acting as nuclear localization signals and/or required for Paf1C and β-catenin binding [2,6,13,15].

Most documented *CDC73* mutations are nonsense mutations that result in physiologically inactive proteins [1,2]. All of them are located in the N-terminal region of CDC73, supporting the functional importance of this region [1,10,16]. Limited proteolytic analysis has shown that the conserved CDC73 N-terminal 111 residues form a globular domain (CDC73-NTD) [1,17]. In addition, CDC73-NTDs contain extended hydrophobic grooves on their surface that may be important for function [1].

We previously reported on the stability of *CDC73* transcripts and translation products in a family with a c.238-1G > A (IVS2-1G > A) splice mutation located in the N-terminal portion of CDC73 encountered in HPT-JT syndrome [18,19]. The *CDC73* transcripts resulting from the splice site mutation were stable, but their translation products were rapidly degraded [19]. Random degradation sites, lysosomes and highly selective proteasomes provide the most important pathways for proteolysis [20,21,22,23,24]. Among these, the ubiquitin–proteasome degradation system is now thought to provide a major mechanism of protein degradation [24].

Since most proteins are targeted for ubiquitin–proteasome degradation [21,22,23,25,26,27], we hypothesized that CDC73 will also be a target of ubiquitin–proteasome degradation. However, there was no evidence that CDC73 was regulated by the ubiquitin–proteasome pathway.

We therefore determined whether CDC73 was ubiquitinated and whether any DUBs interacted with CDC73. Having identified a candidate DUB, we went on to study the biochemical interaction between this DUB and CDC73, to identify the binding domains where these proteins interact, and whether the two proteins co-exist in cells.

## 2. Results

### 2.1. Expression of CDC73 and Polyubiquitination

After transfecting COS7 cells with CDC73 expression vectors harboring FLAG-tagged wild-type (WT) and mutant types (M1, M2) (Figure 1A), the expression of each FLAG-CDC73 mRNA was analyzed by RT-PCR, and expression of the corresponding proteins was examined by Western blotting. Substantial transcription of each mRNA was observed, but only the wild-type protein was detected (Figure 1B). To detect ubiquitination of CDC73, co-immunoprecipitation (co-IP) was performed using an anti-FLAG antibody, followed by Western blotting using an anti-HA antibody. The co-IP analysis revealed polyubiquitination in all three cases, but it was more extensive in the mutants (Figure 1C).

### 2.2. Screening of CDC73-Interacting Proteins

To identify CDC73-interacting DUB proteins, a DUB cDNA library was cloned containing ubiquitin-specific protease (USP) cDNAs, and a yeast two-hybrid screen was performed using this library. Co-expression of a Gal4 DNA binding domain-USP37 construct (as well as similar USP12, USP15, USP25 and USP28 constructs) and a Gal4 activation domain-CDC73 construct in the yeast strain AH109 resulted in blue colonies if interaction occurred. This identified five DUBs out of a family of 58 USPs. USP37 gave the strongest blue color of the five DUBs (data not shown). Glutathione S-transferase (GST) pull-down experiments on the five DUBs confirmed that USP37 was the protein that interacted most strongly with CDC73 (Figure 2B,C).

### 2.3. CDC73 Binds Directly to USP37

To confirm the binding between USP37 and CDC73 shown in the yeast two-hybrid screen, a GST pull-down and co-IP analysis was performed. This revealed that purified GST-USP37 bound to FLAG-CDC73 (Figure 2B) and purified GST-CDC73 bound to Myc-USP37 (Figure 2C) in *E. coli* strain BL21. Reciprocal binding between CDC73 and USP37 was also detected in co-IP experiments performed in COS7 cells using FLAG-CDC73 and Myc-USP37 (Figure 3B). We conclude that CDC73 binds directly to USP37.

### 2.4. CDC73 Is Deubiquitinated by USP37

We next examined whether USP37 deconjugates ubiquitin from CDC73 protein. The level of ubiquitination of CDC73 in COS7 cells was detected using anti-HA antibody and CDC73 was found to be deubiquitinated by USP37 but not by inactive USP37^C350S^ in which the catalytic cysteine 350 was replaced by Serine. Then, the deubiquitination of CDC73 was analyzed using antibodies specific for K48-linked ubiquitin chains, and the specific DUB activity of USP37 on the ubiquitin chain of CDC73 was investigated. This showed that CDC73 was deubiquitinated by K48-specific ubiquitin chains via USP37 (Figure 4A). When the protein synthesis was blocked with cycloheximide (50 μg/mL), expression of CDC73 was found to be increased by USP37 but not by USP37^C350S^ (Figure 4B). We conclude that USP37 deubiquitinates CDC73 via K48-specific polyubiquitin chains, thus inhibiting CDC73 proteolysis.

### 2.5. The Interaction Domains of CDC73 and USP37

We investigated the domain of CDC73 that binds USP37, and vice versa. To locate the exact binding sites, deletions of USP37 and CDC73 were generated (Figure 5A), and co-IP assays were performed between the full-length and deletion constructs. Full-length CDC73 bound to full-length USP37 and its Δ3 and Δ4 derivatives containing ubiquitin-interacting motifs (UIMs) 2 and 3 (Figure 5B). Similarly, full-length USP37 bound to full-length CDC73 and its Δ1, Δ2 and Δ5 derivatives containing the β-catenin binding region (Figure 5C). These results indicate that the interaction between the two proteins requires the β-catenin binding region of CDC73 and the UIM2 and 3 region of USP37.

### 2.6. Co-Localization of CDC73 with USP37

To see if CDC73 and USP37 colocalized in the cell, FLAG-CDC73 and Myc-USP37 vectors were simultaneously transfected into COS7 cells. After co-transfection of FLAG-CDC73 and Myc-USP37 expression vectors in COS7 cells, expression of CDC73 and USP37 proteins, respectively, was observed, thus showing that CDC73 and USP37 were co-expressed at the same location in the cell nucleus (Figure 5D).

## 3. Discussion

This study was conducted to determine whether the CDC73 protein fragment, which was prematurely terminated due to a splice mutation within the CDC73-NTD, would be eliminated via the ubiquitin proteasome pathway. We have shown above that CDC73 is polyubiquitinated via K48-linked ubiquitin chains that are deubiquitinated by USP37, and that CDC73 directly binds to USP37, via an interaction between the β-catenin binding region of CDC73 and the ubiquitin-interacting motifs (UIM2 and 3) of USP37. Moreover, CDC73 and USP37 proteins co-exist in the cell nucleus. We therefore propose that USP37 is a specific DUB involved in the stability of CDC73 in HPT-JT syndrome.

In previous studies, expression of transcripts of a splice mutation in the CDC73 gene in HPT-JT syndrome was detected by real-time PCR in parathyroid carcinoma tissue, whereas the corresponding CDC73 protein was not detected [18,19], and it was shown that the mutant CDC73 mRNA transcribed from the splice mutation was stable in tissue cells, whereas the protein product was rapidly degraded [19]. This suggested that CDC73 might function as a tumor suppressor, consistent with previous findings [1,3,6,7]. However, research on the mechanism by which mRNA transcribed from the splice mutant was lost after translation was lacking.

Over 75% of CDC73 mutations are frameshift and nonsense mutations, which are predicted to result in absence of the translated protein due to nonsense-mediated mRNA decay (NMD) [2,28,29]. NMD is a translation-coupled mechanism that eliminates mRNAs containing premature translation–termination codons and is also linked to pre-mRNA splicing [30]. Aberrant termination induces both translational repression and an increased susceptibility of the mRNA to multiple ribonucleases [29]. The ubiquitin–proteasome and autophagy-lysosomal pathways are the two major pathways of protein and organelle clearance in eukaryotic cells [20,22,23,24]. The ubiquitin–proteasome system acts on the breakdown of misfolded proteins in the endoplasmic reticulum [20,23], and most proteins are targeted for ubiquitin–proteasome degradation [21,22,23,25,26,27].

To prove that CDC73 is also a target of ubiquitin–proteasome degradation, a vector expressing FLAG-CDC73 was constructed and then transfected into COS7 cells. As shown in Figure 1B, CDC73 mRNA was expressed well from both wild-type (WT) and mutant (M1, M2) expression vectors. However, CDC73 protein was only observed in the case of the WT CDC73, not from the M1 and M2 mutants. Both M1 and M2 are truncated proteins resulting from stop codons at the start of exon 4 of CDC73 [18,19], and our data suggest that these truncated proteins are more heavily ubiquitinated than the wild-type protein and thus disappear more rapidly (Figure 1C).

To confirm the presence of DUBs interacting with CDC73, we performed a yeast two-hybrid screening. Five USPs out of 58 USPs were identified by the yeast two-hybrid system, and GST pull-down analysis confirmed that the DUB of CDC73 was USP37 (Figure 2). Repeated GST pull-down analysis using GST-USP37/pGEX-4T-1 and GST-CDC73/pGEX-4T-1 confirmed a direct interaction between USP37 and CDC73. Purified USP37 interacted with CDC73, and vice versa (Figure 2B,C). In co-IP assays using the FLAG antibody, the CDC73 mutants did not interact with USP37, whereas wild-type CDC73 did (Figure 3B). Therefore, it is presumed that CDC73 binds directly to USP37. We then investigated whether USP37 deconjugates ubiquitin from CDC73 protein, and found that ubiquitin bound to CDC73 was removed by USP37, not by USP37^C350S^ (Figure 4A). To reveal the specific DUB activity of USP37, we performed a deubiquitination analysis of CDC73 using K48-linked specific ubiquitin antibodies. This showed that K48-specific polyubiquitin chains were removed by USP37 (Figure 4A) suggesting that proteasomal degradation of CDC73 is inhibited by USP37. We propose that CDC73 is a target molecule of the ubiquitin–proteasome pathway and that USP37 is a DUB protein that interacts with CDC73 protein. Unlike the catalytically inactive form of USP37^C350S^, overexpression of USP37 increased the expression level of CDC73 (Figure 4B). We also propose that USP37 plays a role in stabilizing the CDC73 protein by preventing ubiquitin-induced CDC73 loss.

Finally, to identify the reciprocal binding domains of CDC73 and USP37, we created deletion constructs for both USP37 and CDC73 (Figure 5A). As a result, it was confirmed that full-length CDC73 binds to full-length USP37 and Δ3 and Δ4 that contain UIM2 and 3 (Figure 5B). Conversely, full-length USP37 was found to interact more efficiently with the full-length CDC73, Δ1, Δ2 and Δ5, that contain the β-catenin binding region of CDC73 (Figure 5C). Thus, it appears that UIM2 and 3 of USP37 interact with the β-catenin binding domain of CDC73, and the two enzymes co-exist within the nucleus of COS7 cells (Figure 5D).

The human genome encodes approximately 100 DUBs, of which the USP family is the largest and consists of 58 members [31,32,33,34,35,36]. Three (USP25, USP28 and USP37) of the 58 harbor UIMs that bind ubiquitin [31,36]. However, the roles of the UIMs in these USPs are unknown.

USP37 plays a role in regulating DNA damage repair during mitosis and has three UIMs within the catalytic DUB domain, and the third UIM recognizes the proximal ubiquitin moiety of the K48 di-ubiquitin to enhance cleavage activity [31]. In vitro, USP37 cleaves K48-linked ubiquitin chains, a major ubiquitin chain type that directs target proteins for proteasomal degradation [36,37]. Mutations in UIM2 and/or UIM3 perturb USP37 binding to endogenous ubiquitin–protein conjugates [31,36,38]. UIMs of USP37 contribute to overall enzymatic activity [36], and UIM2 and UIM3 have been suggested to play an important role in promoting the DUB activity of USP37 [31].

This study has some limitations. First, we did not examine the loss of CDC73 mRNA due to NMD and loss of the protein via lysosomal activity. We also did not estimate the binding constant for the interaction between the two proteins or the secondary and tertiary structure of the entire CDC73 polypeptide chain, or test whether the UIM2 and/or UIM3 mutations affect the binding of CDC73. However, this study is novel in showing that CDC73 is ubiquitinated and that its ubiquitination is inhibited by USP37.

## 4. Materials and Methods

### 4.1. Cell Culture, Transfection, and Antibodies

COS7 cells (monkey kidney fibroblast cell line, Korean Cell Line Bank #21651, Seoul, Korea) were maintained in Dulbecco’s Modified Eagle’s Medium (DMEM, Gibco BRL, Carlsbad, CA, USA) supplemented with 10% fetal bovine serum (FBS, Gibco BRL) and 1% penicillin-streptomycin (Gibco BRL). Cells were transfected with an electroporation device (Neon Transfection System, Invitrogen, Carlsbad, CA, USA) according to the manufacturer’s instructions. Parameters were set at 950 pulse voltage and 30 pulse width, and 2 pulses were used. Mouse anti-c-Myc (9E10), rabbit anti-c-Myc (A-14), rabbit anti-HA (Y-11), mouse anti-GST (B-14) and mouse anti-parafibromin (2H1) antibodies were purchased from Santa Cruz Biotechnology (Dallas, TX, USA). Mouse anti-FLAG^®^ M2 and rabbit anti-FLAG^®^ antibodies were purchased from Sigma-Aldrich (Louis, MO, USA). Rabbit anti-CDC73 (D38E12), rabbit anti-K48 linkage-specific polyubiquitin (D9D5) antibodies were purchased from Cell Signal Technology (Danvers, MA, USA). Mouse anti-β-actin and rabbit anti-USP37 antibodies were purchased from Abcam (Cambridge, UK). Goat anti-Mouse IgG (Alexa Fluor 488) and goat anti-rabbit IgG (Alexa Fluor 594) antibodies were purchased from Invitrogen (Carlsbad, CA, USA).

### 4.2. Construction of Expression Plasmids

For transfection and co-immunoprecipitation (co-IP), full-length wild-type CDC73 cDNA and two mutant CDC73 cDNAs (M1 and M2 with 23 bp and 70 bp deletions, respectively) were cloned into pcDNA3.1 vector with an N-terminal FLAG tag (Figure 1A), kindly donated by Hyang Sook Lim (Catholic University, Seoul, Korea).

For yeast two-hybrid screening, Matchmaker GAL4 Two-Hybrid System 3 (Clontech, Mountain View, CA, USA) was used to detect binding between CDC73 and USPs. A full-length cDNA encoding wild-type CDC73 (the prey) was cloned into the pGAD424 vector containing the GAL4 activating domain (Clontech) and full-length cDNAs encoding 58 USP family members (baits) were cloned into pGBT9 vector containing the GAL4 DNA-binding domain (Clontech). For GST pull-down analysis, full-length cDNAs encoding GST fusions of USP12, USP15, USP25, USP28 and USP37 selected by yeast two-hybrid screening were cloned into pGEX-4T-1 vector (Addgene, Watertown, MA, USA) (Figure 2A). A control vector expressing only GST was also constructed. Then, a vector expressing CDC73 with or without GST was constructed by cloning the full-length wild-type CDC73 into pGEX-4T-1. Full-length USP37 (or USP37^C350S^) was cloned into pcDNA3.1 vector with an N-terminal c-Myc tag (Figure 3A). The HA-ubiquitin vector was kindly donated by Jaewhan Song (Yonsei University, Seoul, Korea). To determine the exact binding domains of CDC73 and USP37, deletion constructs of CDC73 and USP37 were generated (Figure 5A). Restriction enzyme sites were created using a QuikChange Site-Directed Mutagenesis Kit (Agilent, Santa Clara, CA, USA) to generate deletions from FLAG-CDC73 WT/pcDNA3.1 and Myc-USP37/pcDNA3.1 vectors. Then the vectors were cloned using an EZ-Fusion™ Cloning Kit (Enzynomics, Daejeon, Korea). The enzyme sites were restored again to the original sequences.

### 4.3. Yeast Two-Hybrid Screening

Strain AH109 yeast cells were transformed with pGBT9-BD-USP vector and incubated on SD plates without tryptophan at 30 °C. After 4 days, resulting colonies were transformed with pGAD424-AD-CDC73 vector then incubated on a nutrient deficiency medium SD plate without tryptophan and leucine containing 4 mg/mL X-α-gal (Clontech) at 30 °C for 4 days.

### 4.4. RT-PCR

Total RNA was extracted using a TRI reagent (Molecular research center, Cincinnati, OH, USA) and cDNA was synthesized using the GoScript™ Reverse Transcription System (Promega, Madison, WI, USA). RT-PCR was carried out using a T100™ Thermal Cycler (Bio-Rad, Hercules, CA, USA). The RT-PCR conditions were 94 °C for 2 min, followed by 35 cycles at 95 °C for 10 s, 58 °C for 30 s, 68 °C for 30 s and 72 °C for 5 min for final extension. Sequences of the primers were as follows: CDC73, 5′-GATTACAAGGACGACGATGAC-3′ and 5′-GACCTATTTCTAAGGGAGCGC-3′; β-actin, 5′- CCATCTATGAGGGGTATG-3′ and 5′- GTAGCACAGCTTCTCCTT-3′.

### 4.5. GST Pull-Down Assay

GST-USP37/pGEX-4T-1 and GST-CDC73/pGEX-4T-1 vectors (Figure 2A) were transformed into *E. coli* strain BL21 and incubated with shaking at 37 °C. A transformant was induced with 100 uM isopropyl β-D-1-thiogalactopyranoside (IPTG, Amresco, Radnor, PA, USA) at 37 °C for 3 h and the cells were sonicated in phosphate-buffered saline with 1 mM phenylmethylsulfonyl fluoride, 1 mM dithiothreitol, proteinase inhibitors, 1% Triton X-100. Purified GST-USP37 or GST-CDC73 was mixed with glutathione Sepharose 4B (GE Healthcare, Chicago, IL, USA) at 4 °C for 1 h. FLAG-CDC73/pcDNA3.1 and Myc-USP37/pcDNA3.1 vectors (Figure 1A and Figure 3A, respectively) were over-expressed in COS7 cells, and the cells were extracted in NP-40 lysis buffer (150 mM NaCl, 1% NP-40, 50 mM Tris pH 8.0) with a protease inhibitor cocktail. GST-USP37 or GST-CDC73 beads and total lysate were incubated at 4 °C overnight and washed with lysis buffer then suspended in SDS sample buffer. Western blotting was performed using anti-FLAG or anti-USP37, and anti-GST antibody.

### 4.6. Co-Immunoprecipitation (co-IP) and Western Blotting

For ubiquitination by overexpression of constructs, FLAG-(WT, M1, M2) CDC73/pcDNA3.1 (Figure 1A) and HA-ubiquitin were transfected into COS7 cells by electroporation (Neon transfection system). For deubiquitination, FLAG-(WT, M1, M2) CDC73/pcDNA3.1, HA-ubiquitin, Myc-USP37/pcDNA3.1 and Myc-USP37^C350S^/pcDNA3.1 (Figure 3A) were transfected into COS7 cells. To identify the exact interaction sites, full-length CDC73 and deletion constructs of USP37 and vice versa (Figure 5A), were transfected into COS7 cells. After 24 h, the cells were treated with 25 μM MG132 (a proteasome inhibitor; A.G. Scientific, Inc., San Diego, CA, USA) at 37 °C for 4 h, extracted with NP-40 lysis buffer with a protease inhibitor cocktail and pre-cleaned with GammaBind G Sepharose (GE Healthcare) at 4 °C for 1 h. The lysates were incubated with anti-FLAG antibody or anti-c-Myc antibody overnight and Sepharose beads were added for 2 h. The precipitated proteins were loaded and separated by SDS-PAGE and transferred onto Nitrocellulose (NC) Blotting Membranes (GE Healthcare). Primary antibodies were incubated at 4 °C overnight and probed with secondary antibodies for 1 h. Signals were detected using Amersham ECL select detection reagent (GE Healthcare).

### 4.7. Immunofluorescence Assay

For co-localization, FLAG-CDC73/pcDNA3.1 and Myc-USP37/pcDNA3.1 vectors were transfected into COS7 cells by electroporation (NEON transfection system). Transfected COS7 cells were transferred into 12-well plates containing sterilized glass coverslips. After 24 h, cells were fixed in 4% paraformaldehyde and treated with phosphate-buffered saline containing 0.2% Triton X-100 for permeabilization. After incubation in a blocking buffer containing bovine serum albumin, cells were incubated with mouse anti-parafibromin antibody and rabbit anti-USP37 antibody. Bound antibody was stained with goat anti-mouse IgG (Alexa Fluor 488) antibody and goat anti-Rabbit IgG (Alexa Fluor 594) antibody. Coverslips were mounted with 4′,6-diamidino-2-phenylindole (DAPI) (ImmunoBioScience Corp., Mukilteo, WA, USA) for nuclear visualization. Fluorescence images were obtained with a confocal laser scanning microscope (LSM800 w/Airyscan, Carl Zeiss, Oberkochen, Germany).

### 4.8. Statistical Analysis

Each experiment was repeated three times, and the results are expressed as mean ± standard deviation of the mean (SD). Statistical significance was tested by one-way ANOVA and set at *p* < 0.05. All analyses were performed using SPSS for Windows (ver. 12.0, Chicago, IL, USA).

## 5. Conclusions

We conclude that USP37 is a DUB for CDC73 and that the two proteins interact via the β-catenin binding region of CDC73 and the UIM2 and 3 region of USP37, suggesting that USP37 is responsible for the stability of CDC73 in HPT-JT syndrome.

## Figures and Tables

**Figure 1 ijms-23-06364-f001:**
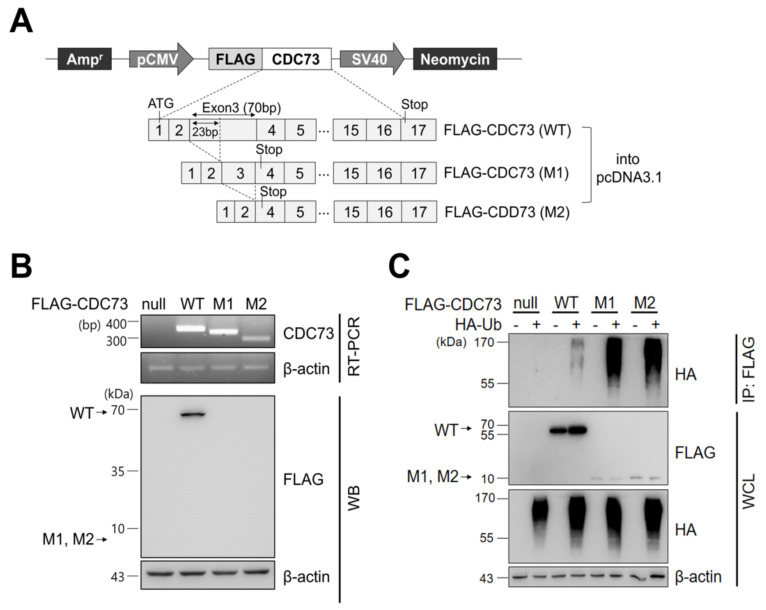
CDC73 expression and ubiquitination. Schematic representation of CDC73 expression plasmids (**A**). For expression or immunoprecipitation (IP), full-length wild-type CDC73 and two mutant CDC73 cDNAs (23 bp and 70 bp deletions, respectively) were cloned into pcDNA3.1 vector with an N-terminal FLAG tag. RT-PCR and WB results of cells transfected with CDC73 wild-type (WT) or mutant (M1, M2) expression vectors (**B**). Transcription of all three mRNAs was observed, but only the wild-type CDC73 protein was detected. To measure polyubiquitination of CDC73 (**C**), co-transfected cells were treated with 25 μM MG132 for 4 h and IP was performed with anti-FLAG antibody. A ladder representing polyubiquitinated CDC73 can be seen in the co-transfected cells. WT, wild-type CDC73; M1, 23-bp deletion; M2, 70-bp deletion; RT-PCR, reverse transcription polymerase chain reaction; WB, western blotting; HA, hemagglutinin; Ub, ubiquitin; IP, immunoprecipitation; WCL, whole cell lysate.

**Figure 2 ijms-23-06364-f002:**
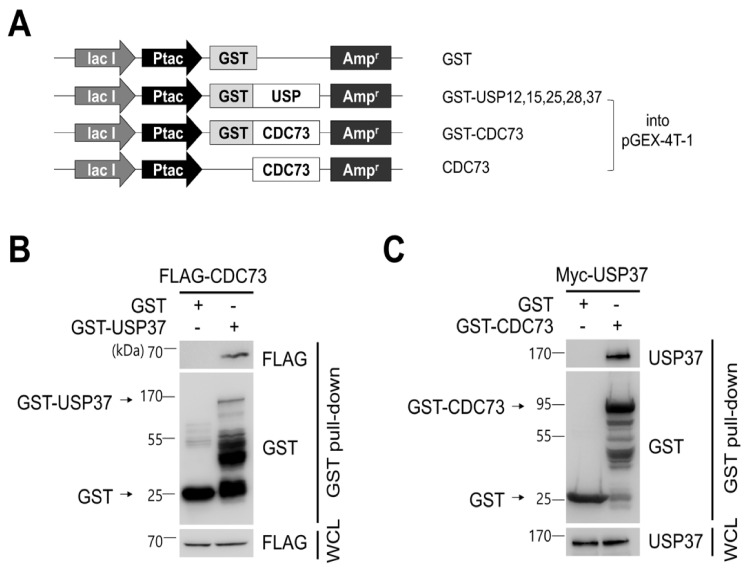
CDC73 binds directly to USP37. Schematic diagram of USPs and CDC73 expression vectors (**A**). For GST pull-down analysis, full-length cDNAs encoding USP37, as well as USP12, USP15, USP25 and USP28, were cloned into pGEX-4T-1 vector. Full-length wild-type CDC73 was cloned into pGEX-4T-1, and a vector expressing only CDC73 without GST was also constructed. GST/pGEX-4T-1 was used as control. GST pull-down analysis using GST-USP37 and GST-CDC73 (**B**,**C**). *E.coli* BL21 was transformed with GST-USP37/pGEX-4T-1 or GST-CDC73/pGEX-4T-1 and induced with 100 uM IPTG for 3 h. Purified GST-USP37 was incubated with a lysate of FLAG-CDC73-transfected cells (**B**), and purified GST-CDC73 was incubated with a lysate of Myc-USP37-transfected cells (**C**). These results show that USP37 is observed to interact strongly with the CDC73 protein. GST, glutathione S-transferase; WCL, whole cell lysate.

**Figure 3 ijms-23-06364-f003:**
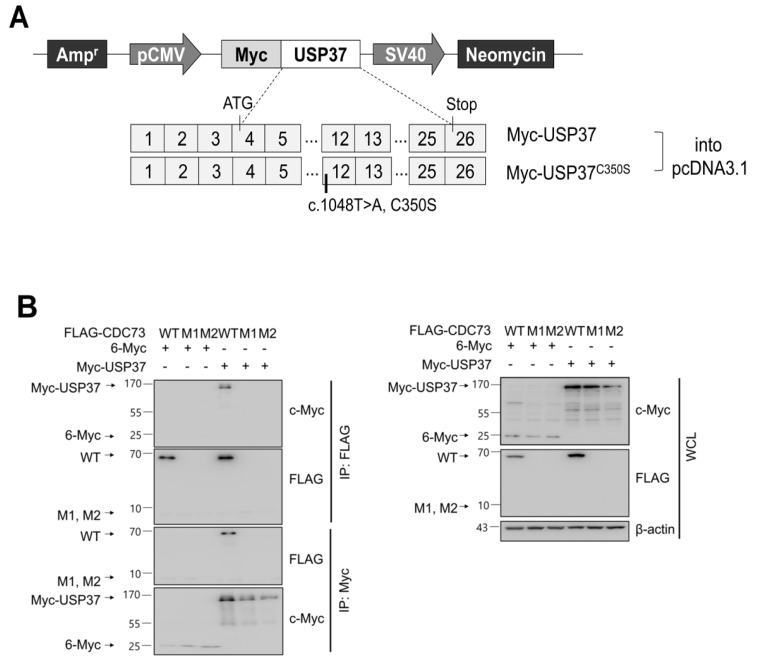
CDC73 interacts with USP37. Schematic diagram of USP37 and inactive USP37^C350S^ expression vectors (**A**). For CDC73 deubiquitination by USP37, full-length USP37 (or USP37^C350S^) was cloned into pcDNA3.1 vector with an N-terminal c-Myc tag. Reciprocal binding between CDC73 and USP37 results from co-IP experiments performed in COS7 cells (**B**). COS7 cells were transfected with FLAG-CDC73/pcDNA3.1 or Myc-USP37/pcDNA3.1 expression vectors. The interaction with CDC73 was tested by co-IP performed with anti-FLAG antibody and anti-c-Myc antibody. Only FLAG-tagged wild-type CDC73 was detected by co-IP with Myc-USP37. These results show that USP37 interacts with CDC73. 6-Myc/pcDNA3.1 was used as control. WT, wild-type CDC73; M1, 23-bp deletion; M2, 70-bp deletion; IP, immunoprecipitation; WCL, whole cell lysate.

**Figure 4 ijms-23-06364-f004:**
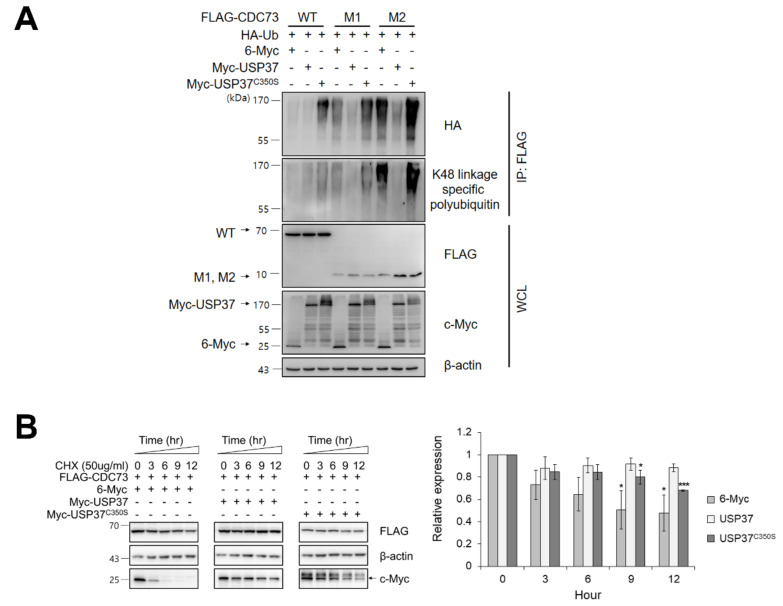
USP37 deubiquitinates and stabilizes CDC73. USP37 deubiquitinates CDC73 (**A**). To examine deubiquitination of CDC73, COS7 cells were transfected with FLAG-(WT, M1, M2) CDC73, HA-ubiquitin, Myc-USP37 or Myc-USP37^C350S^ and 6-Myc and treated 25 μM MG132 for 4 h. Cell lysates were immunoprecipitated with anti-FLAG antibody and Western blotting was performed with anti-HA antibody and antibodies specific for K48-linked polyubiquitin. USP37 stabilizes CDC73 (**B**). To examine the stability of CDC73 expression, FLAG-CDC73 co-transfected along with 6-Myc, Myc-USP37 or Myc-USP37^C350S^ into COS7 cells and the cells were treated with 50 μg/mL cycloheximide (*n* = 3 for each group). * *p* < 0.05, *** *p* < 0.001 vs. 0 h. IP, immunoprecipitation; WCL, whole cell lysate; CHX, cycloheximide.

**Figure 5 ijms-23-06364-f005:**
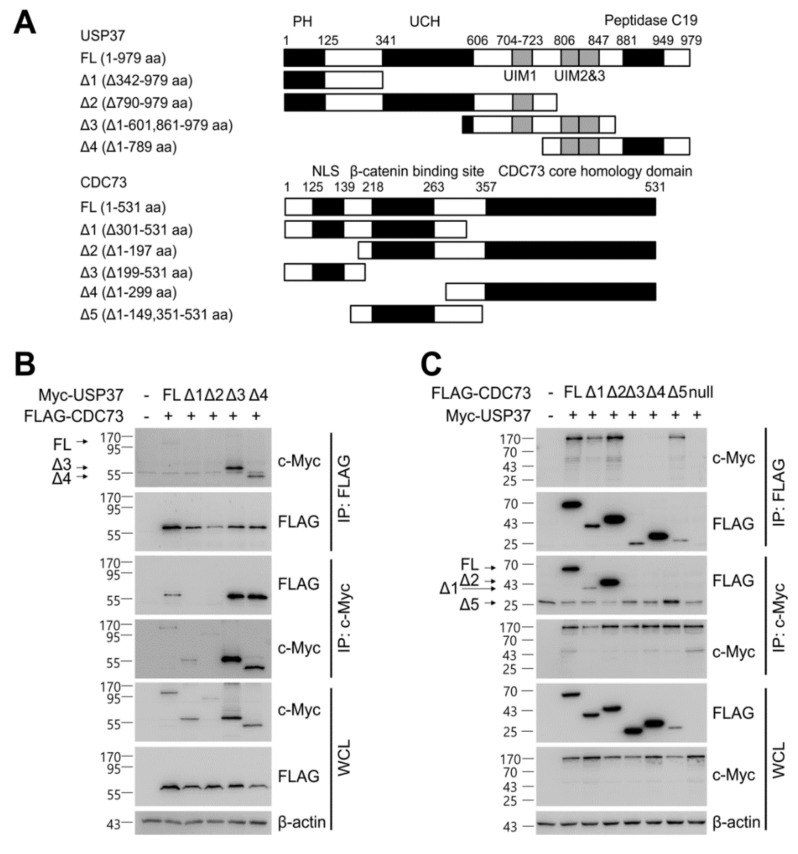
Identification of the binding sites in CDC73 and USP37. Schematic description of the USP37 and CDC73 deletion constructs (**A**). Western blotting results after immunoprecipitation of lysates of COS7 cells transfected with full-length FLAG-CDC73 and Myc-full-length USP37 and deletion constructs (**B**) and vice versa (**C**). Full-length CDC73 binds to full-length USP37, and USP37 Δ3 and Δ4 containing ubiquitin-interacting motifs (UIM) 2 and 3. Conversely, full-length USP37 binds to full-length CDC73, and Δ1, Δ2 and Δ5 containing the β-catenin binding region. Nuclear co-localization of CDC73 and USP37 is shown by the yellow (green + red) nuclear immunofluorescence staining in lysates of COS7 cells transfected with full-length FLAG-CDC73 and Myc-USP37 (**D**). PH; pleckstrin homology; UCH, ubiquitin carboxyl-terminal hydrolase; UIM, ubiquitin-interacting motif; NLS; nuclear localization signal; IP, immunoprecipitation; WCL, whole cell lysate; DAPI, 4′,6-diamidino-2′-phenylindole dihydrochloride.

## Data Availability

Not applicable.

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
