# Peer review of "USP37 Deubiquitinates CDC73 in HPT-JT Syndrome"

_ijms, 2022, doi:10.3390/ijms23126364_

Round 1

Reviewer 1 Report

Building on previous work on medically highly relevant mutations of CDC73, the authors employ cell biology, biochemistry, and imaging to investigate which degradation pathway removes truncated CDC73. They find that CDC73 mutants are degraded by the proteasome and deubiquitinase USP37 stabilizes WT CDC73 by removing K48-linked ubiquitin chains. Investigating further, they confirm that CDC73 stability is affected by the enzymatic function of USP37 and identify the interacting domains. These results are novel and highly relevant and will contribute to understanding the molecular etiology of parathyroid tumors.

The experiments are carried out with adequate methodology, relying exclusively on qualitative approaches. Sufficient controls are presented and original images (blots and light microscopy) show no obvious aberrations. What weakens the presented conclusions is that the interpretation of some experiments is not entirely clear (see “major issues”) and that in general no attempt at quantification of results is made.

The paper is well written and does not need additional language editing. The figures are clear and sufficiently labeled.

If the issues listed below can be addressed, this study fully deserves publication.

Major issues:

Expression of CDC73 mutants: The authors present in figure 2A evidence that WT as well as mutants M1 and M2 are transcribed. They transfect mammalian cells with vectors encoding an N-terminally FLAG tagged CDC73 or mutants of it and find no evidence of expressed CDC73 mutant constructs in an anti-FLAG western blot. They treat the cells with a proteasome inhibitor to stop degradation via the ubiquitin-proteasome pathway and pull-down WT and M1/2 CDC73 using the FLAG tag. They find large amounts of ubiquitin chains for M1 and M2 and less for WT when probing for HA with which ubiquitin is labeled. (Where WT, M1 and M2 would be located on this gel is not clear, I assume that it is similar to the locations on the WCL panel. It would be good to indicate these positions, especially if they are different.) A FLAG western blot of whole cell lysate however shows only WT and no trace of M1 or M2. If degradation of M1 and M2 is stopped and massive signal for ubiquitin chains is registered, why is there no signal for M1 or M2 in the FLAG western blot?

USP37 regulation of CDC73 degradation: To clarify that USP37 deubiquitinates CDC73 in vivo, the authors test whether expression of USP37 or an active site dead mutant (ASM) influences protein abundance and ubiquitination. These experiments, shown in Figure 4, have several issues: 1.) As before, despite inhibition of the proteasome, M1 and M2 are not visible in the FLAG blot. 2.) Expression of WT USP37 is much weaker than expression of the ASM USP37. Could this be adjusted, by e.g. adjusting the transfection protocol? 3.) Ubiquitination of CDC73 WT and mutants is substantially increased when the ASM USP37 is expressed, beyond what is seen without additional expression of USP37. Why? 4.) The time courses shown in Figure 4B show degradation of the Myc tag control and, to a lesser extent, of both USP37 constructs, but the data shown does not show substantial degradation of CDC73. I realize that there are slight differences in band intensity, but when the gel-to-gel variation is not known (n=1, I suppose), it is very hard to draw conclusions from this minor difference. It would be more convincing if an attempt was made to quantify the intensity of the bands and repeat the experiment several times to be able to statistically confirm the difference.

CDC73-USP37 interaction: The authors show interaction between USP37 and CDC73 by co-IP. In addition to WT CDC73, they include the two mutant constructs M1 and M2. There is evidence for expression of neither construct as the FLAG blot for M1 and M2 never shows a band neither in WCL nor in the IPs. The authors conclude in the figure legend (121-122) that “Only .. WT CDC73 was detected by co-IP..”. While this statement is technically correct, it is a bit misleading as the interaction with the mutants could not have been detected in the co-IP assay since they were physically not present to begin with. If no result in the paper shows successful expression of M1 and M2 by FLAG blot, why would they be included in a binding assay?

Minor issues:

Protein degradation concepts: I am not sure the introduction on the history of lysosomal vs proteasomal degradation (62-67) is necessary, as the importance of the ubiquitin proteasome pathway is accepted by the field since decades (reviewed e.g. in PMID: 15688069). I also find that the authors could use at times more precise language, e.g. a protein can be expressed well but be degraded very fast, meaning that it is absent despite being expressed (e.g. 142).

Structural biology: The authors cite the 2017 paper (PMID: 29142233) for the structure of the N-terminus of CDC73, but do not mention the recent structures (e.g. PMID: 34526721) of the entire CDC73 in complex with RNA polymerase, where it is found in direct contact with CTR9 and LEO1. It would be beneficial to include and briefly discuss this work, especially as the beta-catenin binding site is included in the model and conclusions could be drawn with regards to its interactions.

Choice of cell line: Is there a specific reason for using COS7 (kidney cells)?

Identity of ubiquitin chain linkage: This does not require changes to the paper, it’s rather something I find interesting to consider in this context. The authors assert that the ubiquitin chains on CDC73 are linked via K48 (lines 143/144) and while this statement is correct, the experiments presented do not exclude the existence of other linkages on CDC73. Different ubiquitin linkages are used for different signals and while K48 linkage marks proteins for proteasomal degradation, K63 is for example strongly involved in DNA repair signaling. Have the authors probed for other linkages, e.g. K63? USP37 is a K48-specific DUB, but among the other interaction partners identified in this study, some show specificity for other linkages (USP25 -> K11, USP28 -> K63; PMID: 25159004). In the context of this study probing for K48 is sufficient as the main objective is to understand how CDC73 mutants are degraded, but the authors might be interested in looking at other linkages too.

Formatting of figures: The figures do not appear in the order in which they are referenced in the text. Figure 1A-C is a well-made and informative figure, but it should be broken up and added to the panels that show the results so that the construct design is shown directly adjacent to the experimental results, as the authors have done with good result in Figure 5.

RT-PCR: What were the control primers? Were they included in the kit?

Typos and language issues:

The paper is written fluently and does not require language editing.

Affiliations: Almost all authors report email addresses that are provided by free web services such as e.g. gmail, which is a bit odd. Would it be better to use official email addresses and possible to add e.g. Orcid identifiers?

24: interacted -> interaction

26: their respective distinct -> specific

138: K48-linked ubiquitin antibodies -> antibodies specific for K48-linked ubiquitin chains

140: this sentence reads awkward. I am not sure whether we can say that a DUB regulates de-ubiquitination, as it is the enzyme that catalyzes the reaction. While some DUBs are constitutively active, others are regulated either directly or by additional subunits that bind them. The results presented establish that USP37 deubiquitinates CDC73, but does not establish whether this enzymatic reaction is regulated in any way. I would suggest to reword this sentence so it does not appear to claim regulated deubiquitination.

143: please reword this sentence

154: layout

313: layout

324: b -> β (Greek special character)

359: Bounded -> Bound

Figure 3B: Are you blotting for USP37 with a specific antibody or for myc?

Reviewer 2 Report

The manuscript entitled "USP37 Deubiquitinates CDC73 in HPT-JT Syndrome." has been submitted as an original article by Kim et al.

The authors conclude that the deubiquitinating enzyme USP37 acts as a DUB for CDC73, and that therefore USP37 regulates the stability of CDC73 in HPT-JT syndrome.

In general, the experiments are sound and the drawn conclusions are supported by the presented data in most cases.

Important points:

1.) The study has the major problem that the CDC73 mutanst versions M1 and M2 are not detected in teh western blot controls. This is important as it represents a cruical control and is required for the proper interpretation of the corresponding experiments. Therefore, the authors should find a way to detect the CDC73 M1 and M2 proteins in their controls.

2.) Figures in the manuscript: Please add a molecuar weight marker to all figures depictng western blot data. 

3.) Supplemental data: Please add a molecuar weight marker to all of the uncropped western blot data. 

Round 2

Reviewer 1 Report

I am grateful for the authors’ detailed response and their effort to include many of the modifications requested.

I would encourage the authors to include the plot they have provided for point 2-4 into the paper and provide p-values.

Unfortunately, in the revised version, the authors have not addressed the main concern: their inability to show that M1 and M2 are indeed present in their sample. The authors present on two occasions (Fig 1B, 4A) western blots for the FLAG tag of material originating from MG132 treated cells. They show a clear band for the full-length protein and a no trace of a band for mutant proteins M1 and M2. The absence of proteins M1 and M2 from the blot is neither discussed in the original nor the revised text, instead proteins M1 and M2 are treated in subsequent analyses as though they were present although there is no proof of their existence in the sample provided. What confounds this problem is that subsequent analyses of ubiquitination rely on pulldown using the FLAG tag. If we believe that it is possible to pull down copious amounts of FLAG-tagged, ubiquitinated M1/M2, meaning the tag is clearly accessible for binding, why would the tag not be able to bind the antibody during blotting? The authors suggest that one reason could be that the amount is too small, yet, in Fig. 1B and 4A we are looking at MG132-treated cells and there is obviously enough material to result in very large ubiquitin chain signals.

Unfortunately, these technical issues have to be solved before the paper can be published. If the FLAG pulldown and subsequent analysis of M1 and M2 really works as stated by the authors, M1 and M2 are modified by large amounts of ubiquitin chains, increasing their molecular weight substantially. These will change their behavior during gel electrophoresis and blotting and might be the reason why they are not detected during the blotting for FLAG. Common remedies to this situation are the choice of a gel and buffer system that allows the separation of large protein complexes, e.g. gradient gels with TA buffer. SDS-PAGE methodology is not described in the manuscript, therefore it’s difficult to give more detailed suggestions. I’ve personally had good experience with the BioRad precast gels, but many other equally suitable systems are commercially available. You also may have to increase the blotting time to ensure these large, polyubiquitinated complexes are really transferred to the membrane. You can stack blotting membranes if you are dealing with a mix of proteins over a wide range of sizes to avoid losing the small ones. An excellent compendium of strategies is provided in this reference: PMID: 26325464.

I find that the paper is not publishable in its current form as the data shown (absence of M1 and M2) and the interpretation (M1 and M2 treated as being present) do not align. I recommend that the authors find a better method to identify M1 and M2 on the western blot. Once this problem is overcome, this novel and insightful study fully deserves to be published.

Author Response

Response to Reviewer 1 Comments

Point 1: I would encourage the authors to include the plot they have provided for point 2-4 into the paper and provide p-values.

Response 1: Added to the manuscript.

Point 2: Unfortunately, in the revised version, the authors have not addressed the main concern: their inability to show that M1 and M2 are indeed present in their sample. The authors present on two occasions (Fig 1B, 4A) western blots for the FLAG tag of material originating from MG132 treated cells. They show a clear band for the full-length protein and a no trace of a band for mutant proteins M1 and M2. The absence of proteins M1 and M2 from the blot is neither discussed in the original nor the revised text, instead proteins M1 and M2 are treated in subsequent analyses as though they were present although there is no proof of their existence in the sample provided. What confounds this problem is that subsequent analyses of ubiquitination rely on pulldown using the FLAG tag. If we believe that it is possible to pull down copious amounts of FLAG-tagged, ubiquitinated M1/M2, meaning the tag is clearly accessible for binding, why would the tag not be able to bind the antibody during blotting? The authors suggest that one reason could be that the amount is too small, yet, in Fig. 1B and 4A we are looking at MG132-treated cells and there is obviously enough material to result in very large ubiquitin chain signals.

Unfortunately, these technical issues have to be solved before the paper can be published. If the FLAG pulldown and subsequent analysis of M1 and M2 really works as stated by the authors, M1 and M2 are modified by large amounts of ubiquitin chains, increasing their molecular weight substantially. These will change their behavior during gel electrophoresis and blotting and might be the reason why they are not detected during the blotting for FLAG. Common remedies to this situation are the choice of a gel and buffer system that allows the separation of large protein complexes, e.g. gradient gels with TA buffer. SDS-PAGE methodology is not described in the manuscript, therefore it’s difficult to give more detailed suggestions. I’ve personally had good experience with the BioRad precast gels, but many other equally suitable systems are commercially available. You also may have to increase the blotting time to ensure these large, polyubiquitinated complexes are really transferred to the membrane. You can stack blotting membranes if you are dealing with a mix of proteins over a wide range of sizes to avoid losing the small ones. An excellent compendium of strategies is provided in this reference: PMID: 26325464.

Response 2: Following your advice, ths experiment was repeated with reference to the PMID:26325464 paper. When the concentration of MG132 was increased to 25uM and transfer was reduced from 60min to 40min, the expression of M1 and M2 could be confirmed. Therefore, figure 1C and 4A were modified.

Reviewer 2 Report

The revised version of the manuscript entitled "USP37 Deubiquitinates CDC73 in HPT-JT Syndrome." has been submitted by Kim et al.

The authors have solved the two easier points that I had raised concerning the first version of the manuscript.

However, the major problem still exists: The authors cannot provide the important control that the mutant forms M1 and M2 are expressed as they cannot detect them in Western Blots. This is a cruical control and this problem needs to be solved in order to fully support the conclusions drawn from the corresponing experiments.

conclude that the deubiquitinating enzyme USP37 acts as a DUB for CDC73, and that therefore USP37 regulates the stability of CDC73 in HPT-JT syndrome.

In general, the experiments are sound and the drawn conclusions are supported by the presented data in most cases.

Important points:

1.) The study has the major problem that the CDC73 mutanst versions M1 and M2 are not detected in teh western blot controls. This is important as it represents a cruical control and is required for the proper interpretation of the corresponding experiments. Therefore, the authors should find a way to detect the CDC73 M1 and M2 proteins in their controls.

Author Response

Response to Reviewer 2 Comments

Point 1: The study has the major problem that the CDC73 mutanst versions M1 and M2 are not detected in teh western blot controls. This is important as it represents a cruical control and is required for the proper interpretation of the corresponding experiments. Therefore, the authors should find a way to detect the CDC73 M1 and M2 proteins in their controls.

Response 1: When the concentration of MG132 was increased to 25uM and the transfer was reduced from 60min to 40min, the expression of M1 and M2 could be confirmed. That is, figure 1C and 4A were modified.

Round 3

Reviewer 2 Report

The authors have submitted a revised version of the manuscript, in which they have solved the previously mentioned problems.